# “You Don’t Look Dyslexic”: Using the Job Demands—Resource Model of Burnout to Explore Employment Experiences of Australian Adults with Dyslexia

**DOI:** 10.3390/ijerph191710719

**Published:** 2022-08-28

**Authors:** Shae Wissell, Leila Karimi, Tanya Serry, Lisa Furlong, Judith Hudson

**Affiliations:** 1School of Psychology and Public Health, La Trobe University, Bundoora, VIC 3086, Australia; 2Psychology Department, School of Applied Health, RMIT University, Melbourne, VIC 3001, Australia; 3School of Medicine and Healthcare Management, Caucasus University, Tbilisi 0102, Georgia; 4School of Education, La Trobe University, Bundoora, VIC 3086, Australia; 5Child Well-Being Research Institute, University of Canterbury, Christchurch 8140, New Zealand; 6School of Education, University Tasmania, Hobart, TAS 7001, Australia

**Keywords:** JD-R Model, dyslexia, workplace, burnout, disability, employment

## Abstract

Background: Employment and job security are key influences on health and wellbeing. In Australia, little is known about the employment lifecycle of adults with dyslexia. Materials and Methods: Using a qualitative research design, this study sought to explore the experiences faced by adults with dyslexia seeking and retaining employment. In-depth interviews were conducted with a cohort either currently or previously in the labour market. We used the Job Demands Resource Model of Burnout (JD-R Model) to explore links between workplace characteristics and employee wellbeing. Deductive content analysis attained condensed and broad descriptions of participants’ workplace experiences. Results: Dyslexic adults (n = 14) participated; majority employed part/full-time and experienced challenges throughout their employment; exhaustion and burnout at work were reported, also fear and indecision about disclosure of dyslexia. A minority reported receiving positive, useful support from team members following disclosure. Conclusion: The JD-R Model provided a guiding framework. We found participants experienced a myriad of challenges that included risk of mental exhaustion, discrimination, limited access to support and fatigue, leaving them vulnerable to job burn-out. Dyslexia does not have to be a major barrier to success in any occupation. Yet, when in supportive, informed workplace environments, employees with dyslexia thrive.

## 1. Background

Gainful employment is a crucial social determinant of health, allowing for individuals to participate meaningfully in society and providing a scaffold for social interaction and economic independence [1]. Extended periods of time in unemployment or underemployment are associated with financial strain, social isolation, and poor mental health and well-being [1,2]. Although there are many factors that contribute to employment difficulties, people with various types of disability are more likely to be unemployed compared to people without disability [1,3]. In Australia, 53 per cent of working-age people with disability were in the labour market compared to 84 per cent of people without disability [4,5]. Australian research by Wissell et al., (2021) identified that working Australian adults with dyslexia had an unemployment rate of 7.7% which is slightly above the current ABS data for the overall population, sitting at 7.1% [6,7].

Dyslexia is a specific type of learning disability associated with poor reading and spelling skills, thought to be underpinned by an underlying phonological processing deficit [8,9,10]. Dyslexia is a lifelong condition [11,12,13] and affects individuals across the IQ range [11]. Evidence suggests that dyslexia occurs in up to one in ten people [11,14,15,16].

Over the last decade, there has been an increased focus on the psychological risks and educational needs of dyslexic individuals who are still at school or in post-secondary education [17,18,19,20]. However, less attention has been focused on the process of dyslexic adults entering the workforce [21]. Additionally, while legislative policies in Australia and overseas have been developed to protect people with dyslexia from discrimination [22,23,24,25,26], a commitment to those with dyslexia in workplaces has not been as well established [24,27].

With advances in technology and digital communication, vocational success increasingly relies on competency in literacy, which can leave those with dyslexia vulnerable in the workplace [1,28,29]. A systematic review by De Beer et al., (2014) examining the workplace experiences of people with dyslexia [30], suggested that adverse experiences could be minimised if employees with dyslexia had: (i) greater autonomy to implement their own workplace adaptations and coping strategies to manage their work, including access to assistive technology, (ii) enhanced self- awareness of their own strengths and difficulties, and (iii) more support from employers and colleagues to disclose their dyslexia and use adaptations at work.

Various authors [1,24,31,32] reinforce the importance of training to be offered to employers and their leadership teams to improve their knowledge of dyslexia. It appears from the research that without a coordinated effort from employers, managers, workplace leaders and human resources staff to provide accommodations and adjustments, employees with dyslexia are likely to struggle to complete designated work tasks [24,30,32]. However, when employers and workplace leaders have insufficient knowledge about dyslexia, employees with dyslexia are more reluctant to disclose their condition, fearful that exposure may result in stigma and discrimination [21,33,34,35,36]. This is unfortunate, given the evidence demonstrating that disclosure can have positive effects on a worker’s health, social relationships, self-identity and job performance, as well as the organisation in which the worker is employed [37,38].

There is scarce theory-driven research and a significant need in Australia to increase understanding regarding aspects in the work environment that enrich or hamper occupational health and satisfaction, from the standpoint of employees with dyslexia. Therefore, in this study, we wished to explore and better understand the working environment for people with dyslexia. To do this we have adopted the Job Demands–Resources Model of Burnout (JD-R Model), proposed by Demerouti et al. [39] as a theory-driven framework.

The Job Demands-Resource Model of Burnout (JD-R Model) (Appendix A) is a validated tool that has been used to evaluate organisational outcomes across several workplace settings [39,40,41,42] and has generic application for all workplaces and their varying composition of employees. Currently, there is only a small body of research using the JD-R Model on populations with disabilities [43,44,45], however not on a disability population such as dyslexia. For those with dyslexia it is not clear if, how and why workplace job resources and job demands can be connected to work-related outcomes (such as job burnout).

The aim of this research was to investigate the work experiences of adults with dyslexia by exploring organisational practices and derive suitable recommendations to support their participation in the workforce. This paper forms part of a larger project investigating the lived experiences and the social and emotional wellbeing of adults with dyslexia in Australia across the domains of education and employment.

## 2. Methods

### 2.1. Design

#### Methodological Framework

As there is little research exploring the associations between workplace demands and employee wellbeing for people with dyslexia, we adopted a qualitative design utilising in-depth interviews to gather substantive responses from participants. We chose semi-structured interviews, utilising an interview guide with a set of pre-determined questions. Further follow-up and clarifying questions emerged during interviews with participants. Qualitative research does not specify a fixed amount of data to be collected but continues until saturation is reached and the available data tell a rich, complex and in-depth story about the phenomenon of interest. Compared to quantitative research, sample sizes for qualitative research are typically much smaller [46,47]. Saturation of data occurs when no new insights can be gained; this is commonly achieved with minimum sample sizes of 12 for qualitative studies [46,47,48,49]. As a result, 14 participants were considered adequate for this qualitative analysis and the scope of this research.

We have not used the JD-R Model to compare whether our population are facing different challenges to the general population. Instead, the aim was to support the development of our categorisation framework and to utilise a theoretical framework to better understand and study the association between job demands and employee wellbeing and motivation, and to explore how job demands may be mitigated by job or personal resources (occupational support/adjustments or personal characteristics/strategies).

### 2.2. Participants

Purposive sampling is generally used for qualitative research as recruitment is based on the shared experience of participants that is of relevance to the research question [46]. Purposive sampling was used to enroll participants who met the subsequent criteria: (1) living in Australia; (2) at least 18 years of age; (3) have received a formal diagnosis of dyslexia and (4) have participated in the workforce for a minimum of three years. Notification of the study was sent via electronic mail to all members of the Dear Dyslexic Foundation (DDF) database and the Equal Employment Opportunity Network Victoria database (a forum dedicated to educating decision makers on creating more diverse, inclusive and equitable workplaces). In addition, various open and closed social media platforms, including LinkedIn, Twitter and Facebook, were used for study recruitment. Fourteen participants were recruited: eight females and six males.

### 2.3. Data Collection

The first author, who is also dyslexic, conducted one semi-structured in-depth interview with each of the 14 participants via Zoom. The 42 interview questions (Appendix B) were generated by the authors, based on previous literature about the profile of adults with dyslexia, e.g., [28,30,50,51,52] and in line with the study’s purpose. Of the 42 questions, 15 questions asked about participants’ dyslexia history and education, but the remaining 27 questions focused on their workplace experiences and their perceived psychological wellbeing in relation to working. Interviews ranged from 24 min to 72 min (mean = 47 min) and were audio-recorded for later transcription and analysis. At the beginning of each interview, the lead investigator disclosed her own dyslexia to participants. This was intentional since disclosure can enhance a researcher’s vulnerability [53], and enable authenticity and transparency to be established early on, and provide greater context for participants of the research [53,54,55]. The interviewer kept field notes and conducted reflective audio recordings following each interview to assist in building a rich understanding of participants’ experiences in the context of the phenomena under review [56,57].

Prior to data analysis, each participant received their verbatim transcript and was invited to review and request changes to any part of the contents within two weeks of receipt. Minor amendments were requested by two of the 14 participants to remove identifiable data. At this point, transcripts were uploaded to NVIVO software (version 12), participants were assigned a number code, and their transcripts were de-identified. Ethics approval was granted from La Trobe University to conduct this study.

### 2.4. Analysis

Deductive content analysis [58] was used to prepare, organise and analyse the interview data. The preparation phase involved familiarisation with the interview data through multiple active readings of the transcripts and listening to audio files.

To organise the data, a categorisation framework was developed based on the six main areas of the interview questions; (1) diagnosis; (2) dyslexia across the life span; (3) dyslexia and support networks; (4) mental health; (5) workplace experiences and (6) workplace improvements). Categories of data across the interview transcripts were identified such as, the major category of *Workplace Experiences*, with the subcategory *Reasonable Adjustments-including text to speech software*, *editing support* and *mobile phones*. The *workplace experiences* and *workplace improvements* categories were interrogated further by searching for frequently recurring words (e.g., *anger*, *frustration*), and coding ideas reflected in utterances (e.g., *no one could make me feel as bad as when I was six years old; I have a team that supports me with day-to-day activities*). This open coding process led to the clusters of content that fell under *workplace experiences* and *workplace improvements* categories. To demonstrate, under *workplace experiences*, the subcluster of content about job demands emerged and included sub-categories such as *not being able to meet the expectations of ‘normal’ people*, *fractures to ‘on-the-job’ working relationships* and *systemic organisational barriers* were created [58].

Finally, all transcripts were re-read for reflection and to ensure that interpretation of participants’ data was captured authentically by the categorisation matrix. Direct quotations from participants were cross-checked against the categories to ensure the categories were reflective of the participants’ narrative [59,60]. While the researchers had predetermined topics for discussion within the interviews, the generic and sub-categories themselves were generated from the interview data using the process described.

### 2.5. Ensuring Rigour

Several steps were implemented to ensure rigour in our data collection and deductive content analysis. Data collection methods were co-constructed by the entire research team to ensure that the interviewer’s personal experience of dyslexia did not introduce potential bias in the interview questions. A second member of the team verified the fidelity of the structured interviews by listening to 10% of the recordings. Three weekly research meetings were held to reflect on the findings, coding processes and analysis to ensure that the interpretation of data was not overlaid with the interviewer’s lived experiences. These meetings were recorded, and an audit trail was kept of all research actions undertaken from the start of the research project through to the reporting of findings, including the steps taken to ensure verification and dependability of findings [61]. The researchers also ensured the transferability of the project by providing detailed characteristics of the participants [61].

## 3. Results

### 3.1. Participant Characteristics

Results are represented in three sections: participant characteristics, workplace experiences as reflected by participants, and their suggested ideas for workplace improvements. Fourteen participants were recruited: six male and eight female. They ranged in age from 20 to 69 but clustered within the ages of 30 to 49 (See Table 1). To protect their privacy, participants will be referred to as ‘they’. Eight participants reported receiving their diagnosis of dyslexia in childhood or adolescence, while the remainder received their diagnosis as an adult, with one participant reportedly being diagnosed in their early fifties. Four participants reported additional co-morbid diagnoses such as auditory processing disorder, dysgraphia, or dyscalculia.

Diagnoses were largely provided by psychologists, although there was some variation. Over half (8 out of 14) of participants reported a past familial history of dyslexia among first-or second-degree relatives and three (21%) also stated that they had children with dyslexia.

In terms of education, all but one participant had completed secondary school in Australia, and all had attained post-school qualifications at various vocational or tertiary levels. More than 93% of participants had completed a post-secondary education, which is notably higher than the national population average of 78% [62].

Participants worked across a variety of settings including service industries, education, public service, and commercial enterprises. The majority (n = 8) worked in full time roles. One participant, previously employed, was actively seeking work at the time of interview, and one was semi-retired.

### 3.2. Pre-Employment

Although our aim was to examine workplace practices, many participants also shared their experiences of seeking work. Therefore, we have also explored perceptions with the process of recruitment. Due to the challenging and somewhat traumatic experiences spontaneously expressed by some participants, and given most participants attributed these difficult situations to their dyslexia, we felt it was important to describe their perspectives of the pre-employment process because of its intimate and essential connection to employment.

Participants (n = 3) were required to complete tasks such as pre-interview aptitude tests or write a report in the pre-employment process and were not provided with any adjustments (such as additional reading time), even when some tried to advocate for their needs. These participants stated in general they were left with a sense that they were not on an even playing field with other candidates during the interview process. For example, Participant 8 (P8) explained that they were asked to write a report on one of the interview questions using a computer that did not have any assistive technology. P8 stated, *“I can’t do this,”* and was told *“Just try your best”*. P9 described being unable to complete an aptitude test and feeling humiliated. They said: *“… I failed the government test. After the test, two ladies sat in front of me and said, ‘You don’t look dumb. We’ve never seen anybody fail a test as bad as this.’ And I said ‘Well, I’m dyslexic’ as my eyes dropped…”* Likewise, many participants felt unable to demonstrate their full capacity and employability to prospective employers and associated this with being unsuccessful in securing certain jobs.

### 3.3. Employment

#### 3.3.1. Workplace Experiences

##### Job Demands

Three sub-categories were identified in relation to job demands. These were: (i) *not being able to meet expectations*, (ii) *fractures to ‘on-the-job’ working relationships*, and (iii) *systemic organisational barriers*. These are discussed below.

Not being able to meet expectations

All participants reported perceiving stigmatising attitudes by managers and colleagues at some point within their careers. Regardless of sector or role, they spoke universally of how they struggled to meet job demands, a struggle they ascribed to having dyslexia. For example, P10 said:


*“… [How do I] manage priorities of conflicting pressures and multiple tasks with the burden [of] writing and editing it? How do I get it done in time with [an] influx of work at the same time? The mental load that this has on someone who struggles to organise thoughts, priorities’, juggle the plethora of parts… It’s an enormous mental load…”*


This was compounded by participants feeling that there was little understanding from many colleagues about what dyslexia is (and is not), leading to feelings of being ‘unheard’.

P13 explained that even after disclosing their dyslexia, *“… the disability was quickly forgotten…”* and workplace literacy demands such as reading large amounts of text, using correct spelling, and producing reports to a certain standard were still presumed and *“…added to stress levels within the workplace…”*

Many participants linked their stress to feelings of anger, frustration, anxiety and for some, feelings of depression. For P10, trying to do a good job at work elicited a complex mix of emotional responses, *“…you don’t want to be judged on it all the time and yet you can’t put in the hours that it takes to make it perfect either, so I just feel so much anxiety. I’m so angry that I have to be so dependent, and I can’t do my own job and I have this extra pressure on me…”*

Finally, some participants also felt their dyslexic struggles had affected their career prospects either within the organisations they worked currently or when seeking new career opportunities. Lack of career progression created frustration for participants who knew they could take on new work given the right support. This was demonstrated by P3 who said, *“…I want to use the big word that’s in my head, but I can’t because I don’t know how to spell it. So, I’ve got to crank it down to a simpler term. It’s probably cost me, promotion wise, a few times…”*

Fractures to ‘on-the-job’ working relationships

Collegial relationships can be integral to managing job demands and collegial workplace environments. Many participants felt relationship building was constrained by their shame about having dyslexia and ironically, some felt somewhat fraudulent when they performed well at work. P6 articulated this when recalling a comment by a colleague, *“…Well, if you can achieve that then there’s not a problem with you. You don’t have a disability…”* Some participants described being the recipient of seemingly innocent but hurtful and demeaning comments. P3 spoke of feeling humiliated but powerless by so-called jokes and said *“…I felt like walking out of a meeting—one guy said, ‘If you’re dyslexic this won’t make any sense to you’, making a joke of it like you’re stupid or something. I don’t really appreciate that. But then you can’t really take it to heart. I mean it’s just a simple joke really…”* P6 was told *“…you don’t look like a dyslexic…”* P13 also described a different experience where their colleagues were well-intentioned and tried to help but were misguided in their efforts. *“One of my colleagues started to speak to me slowly so that I would understand. Another wrote to me in very large font…”*

Participants repeatedly described that colleagues either assumed that having dyslexia meant they were ‘incompetent’ or ‘lazy’. As P11 described, *“…In any form of written communication it’s very hard to proof my work, very hard to see the mistakes I make. Therefore, any mistakes can be contrived to be a factor of laziness or a reflection of not understanding the topic or rushing my work…”*

Systemic organisational barriers

Many participants maintained that at the systemic level, their workplace lacked various inclusive practices that would have assisted them both at the onboarding phase and as employees. As a newly appointed employee, P8, who was required to sign a lengthy contract of employment and given very little time to do so, noted *“… [Dyslexics] are just going to sign off on it regardless and so there needs to be an alternate strategy for them to, at least, comprehend what was in it…”* P13 spoke of the dilemma of whether to tick the box that indicated they had a medical condition/disability *“…I remember the form I had to sign to gain employment, [it] said ‘do you have a medical condition?’. I hesitated—I still remember it, I hesitated, because I wondered whether this would be attributed as a medical condition. I ticked the box that said I didn’t and was gainfully employed for several years…”* Participants expressed that these experiences left them feeling disheartened, devalued and at times humiliated, prior to even commencing in the role.

Although participants were provided with standard workplace documents that met statutory requirements (e.g., anti-discrimination policies and, in some organisations, inclusion and diversity policies), many noted an absence of formal processes and procedures to support those with dyslexia. P12 stated that *“…although there was an inclusion and diversity committee at work, there’s nothing in place at the moment for people with learning disabilities…”* Some felt dyslexia did not attract the same level of awareness and support as other types of diverse needs, such as employees with English as a second language.

Even when individual leaders/managers took the time to understand and provide support to dyslexic employees, a lack of consistent, organisational-wide practices left some participants feeling vulnerable to changes in leadership. P9’s story demonstrates this: *“…I told him [my supervisor] that I was dyslexic. He did research on it and then sat down with me and said ’Okay, well, we’ll work on this together. Feel free to tell me when you’re struggling and things like that.’ Which was really good. But then he left. My next supervisor turned around and said he didn’t care that I was dyslexic. Too bad…”*

The lack of system-wide inclusive practices described by participants meant that for many, their workplace culture became one of concealment rather than authentic embracement of and support for diversity. This was exacerbated by the fact that many participants believed their leaders or colleagues had little awareness of dyslexia and its impact. P11 stated: *“…Sometimes I just needed to work from home because I needed to concentrate. At the moment when I say that, they would think that I’m being a bit precious. They wouldn’t relate that to dyslexia…”*

##### Employee Wellbeing

Employee self-awareness

Most participants had constructed much of their identity around their dyslexia. P1 stated, *“…I think fundamentally it’s [dyslexia that] made me who I am and I love it and I love the strength that it gives me in other areas…”* and P2 said, *“…I learnt to be resilient, to be strong in myself, to build confidence…”* Participants in leadership positions (n = 5), reported that their seniority afforded them the capacity to take advantage of supports which eased the burden of dyslexia on their work. P12 described: *“…My team know I’m [dyslexic] and I need time to read through [documents] before we discuss them. They’ll say, ‘Hang on, why don’t I leave this with you for a little bit and we’ll come back to it.’ They are probably more respectful of how our brains work…”* For some, disclosure helped to build autonomy and control in the workplace. As P10 explained: *“…I needed my team to know [I’m dyslexic] because that’s where my weakness is, it’s in the execution of things and I needed their support to execute, [so I can] then report up to the CEO…”*

Resilience

When participants had a sense of autonomy about how they managed their dyslexia in the workplace, it appeared they were more likely to have strong levels of self-efficacy. P2 stated, *“…I’ve become very strong with strategies and resilience around how to cope and how to achieve—and build on my strengths, and then I’ve learnt probably over the last 10 years, probably 20 years to get people in to help me…”*

Conversely, a failure of some organisations to respond to reasonable requests for workplace adjustments was perceived by some participants as a trigger for feelings of mistrust towards the organisation. One participant noted that in their workplace, *“…it took a long time to do anything…”*, while another commented, *“…They always put on their best face, but when it comes to actioning items, it takes a long time if it is going to happen…”* Some participants believed that if the change or support requested was going to cost money, management were less likely to follow through.

Job burnout

Mental fatigue was described frequently as being the result of excessive cognitive demands. For example, P9 said *“…It takes me more of an effort than most people to do things. I would come in early or stay late or skip lunch and things like that. I had to work extra hard…”* Even when participants took extra time and used various tools to support their writing or spelling, they still did not necessarily feel they met required standards. P3 noted: *“…I use spellcheck … that’s great, and then re-read three, four times, make sure it makes sense. And still you sometimes get an email back and they say, ‘I don’t really understand what you meant’…”*

Additionally, some participants felt that the sensory environment at work was not conducive to their work productivity and compounded mental fatigue. P5 said: *“…I realised I was struggling quite a lot. I wasn’t able to make my normal KPIs. I realised I was getting sensory overload. [My dyslexia] was making me feel quite unhappy, uneasy, making me fear going to work…”* Other participants spoke of choosing to wear headphones in the office, locking themselves away in rooms or working from home (pre COVID-19) to enable them to concentrate on completing tasks. The culmination of feelings such as stress and sensory overload that some participants described, as well as the mental energy required to get through the day, meant that mental fatigue was a significant and widespread problem and a source of frustration for many of our participants.

Despite the commitment to their workplaces, many participants described a sense of chronic stress related to their work. Participants regularly reported experiencing ongoing exhaustion, often leading to low levels of professional self-efficacy including feelings of self-doubt and fear about work. These feelings arose because of the perceived challenges they linked to their dyslexia. P12 illustrates *“…I have such mixed feelings about it [my dyslexia] because on some days […] you see more mistakes from people who don’t have disabilities, [so] why do you care? But I think it’s kind of self-consciousness that you build up. It [dyslexia] contributes to that sense of being overwhelmed [with] self-doubt…”*

Participants commented repeatedly about feeling self-conscious about how long it took to complete tasks, which in turn appeared to generate negative feelings such as paranoia, worry and frustration. Some began to question their own capacity. P5 shared that their dyslexia made them feel *“…quite unhappy, uneasy, making me fear going to work and made me question my own career path as well….”* P11 spoke openly about how her dyslexia mistakes could be *“…contrived to be a factor of laziness, a reflection of not understanding the topic or rushing work. And therefore, I think that does limit my ability to communicate and limits my ability to progress in the work…”* P3 believed their dyslexia had actively limited their career: *“… I get depressed about it sometimes…I can easily say it’s cost me one promotion…”*

Self-disclosure of dyslexia

Self-disclosure was a vexed issue for the participants in this study, with some choosing to disclose and others feeling great reluctance. For those participants that did disclose, some did so during the job interview while others disclosed when they felt the need to justify why there were errors in their work. Two participants explained that they chose to disclose only when they had more freedom in senior roles and felt the need to advocate. P1 stated, *“…I felt it almost an obligation to figure out how to make it known, so that I could fight the very prejudice that I’d seen for so long, by normalising it…”*

Experiencing work as a psychologically unsafe environment also posed barriers to disclosure. This reluctance was largely motivated by a fear of being judged as *different* and consequently being stigmatised. Nevertheless, some noted that their disclosure was met with acceptance but often led to a greater sense of vulnerability, pity, not being taken seriously and being perceived as a weakness. This was articulated by P11 who said *“…I would avoid that [disclosure] because I don’t feel it would have a beneficial outcome. [I would be] exposing myself as [having a] weakness…”* P14 described feeling humiliated when colleagues would find errors such as leaving out words in emails *“… [it was] humorous to them which I really got a bit defensive of and a bit upset about…”*

### 3.4. Workplace Improvements

All the participants typically described a lack of disability-appropriate job resources (the physical, social, or organisational factors that help an individual achieve their work goals and reduce stress) that they need to work at their full capacity at different points in their careers. For example, as noted earlier, P11 described sensing that working from home from time to time (which they felt enhanced their concentration) was not acceptable.

Provisions in workplace budgets for reasonable adjustments to the working environment and access to training or assistive technologies were also urgently recommended. As P1 illustrated *“…I think there’s probably two parts. I think there is a part about helping people see the strengths and what dyslexic people can do and then I think there’s a part about some of those accommodations, like access to proof reading and readily available technological help would be very useful. Making sure those things are just readily available without it being a fuss…”*

Some participants also described what a ‘dyslexic friendly workplace’ could look like. Central to this notion was training for all colleagues, at all levels of the hierarchy, in dyslexia awareness (including disclosure of disability), equal rights, and the use of assistive technology. Learnings would also need to be actively embedded into the workplace.

Several participants believed that the culture of their workplace could be strengthened by simple organisational changes championed by the leadership team in a ‘top-down’ approach. This was highlighted by P13 who said: *“…I believe in the tone at the top being important, that the culture of the organisation is accepting of diversity and equal opportunity….”*

Participants displayed an eagerness to succeed at work and believed that the introduction of formal and informal strategies would create a sense of trust and autonomy within teams benefitting both the employer and employees. P1 discussed:


*“…I’m very lucky. I have one person who looks after the diary and I have others who makes sure I am where I should be with what I need, we’ve got that organised. There is an overall strategy and structure to how we shape the week’s commitments at work and ongoing tactical conversations to keep the day-to-day bit organised. There really isn’t a bit of my day—in fact, not much of my life—that isn’t planned and organised ahead of time, to keep me where I need to be, doing what I need to be doing…”*


## 4. Discussion

This study sought to investigate the lived experiences of working Australians who have dyslexia. Based on the JD-R Model, it has been established that high job demands in the absence of job and personal resources can lead to poor mental health and wellbeing [40,41,63,64,65,66]. We wanted to investigate the workplace practices described by employees with dyslexia and how job demands, job resources and personal resources might be further impacted by dyslexia, and what the result might be on mental health and wellbeing. We also wished to explore whether organisations are fostering inclusive practices to support employees with dyslexia or other learning disabilities, and what more they might seek to do.

Our research highlighted challenges for adults with dyslexia in seeking work, starting and retaining employment, retaining employment and progressing a career. Although we did not anticipate exploring job seeking experiences, it became apparent during data collection that many participants experienced what they perceived to be discrimination, unfair judgments, and a lack of access to reasonable adjustments during screening and recruitment processes. This accords with previous international literature suggesting people with dyslexia are hesitant to disclose their disability during the recruitment process due to fear of discrimination [34,67,68,69], despite legislation which is designed to be protective.

Once employed in a role, people with dyslexia face a myriad of challenges trying to keep up with workplace demands, particularly when their disability was unsupported. Similar to local [70,71,72,73,74,75] and international research [39,40,41,63,76] that has used the JD-R Model we also found that when employees are faced with high job demands and low job resources that they can face high levels of job burnout. However, we could not determine whether those with dyslexia faced additional job burnout because of their disability compared to the general population, or differences in job burnout based on specific industries.

Research undertaken by Lehmann (2021) found that an increase in Multiple Sclerosis (MS) work challenges could lead to complications meeting job requirements and could consequently lead to feelings of increased fatigue and burnout, leaving employers vulnerable to employees with MS leaving their jobs. When MS-related work difficulties were integrated within the health-impairing process, it’s believed that taxing job attributes enhance apparent MS-related work difficulties [45]. In line with Lehmann’s work, we found for our participants that their dyslexia difficulties could lead to burnout and reduction in energy due to the difficulties of meeting work requirements. As dyslexia difficulties are embedded within the health-impairing process, demanding job characteristics are assumed to make dyslexia work more challenging for people. This may indicate that those with disabilities are at greater risk of job burnout and further research is warranted to better understand whether people with dyslexia are experiencing the same workplace difficulties as those within the general population or if their perceived dyslexia difficulties add another layer of complexity leading to exacerbated job burnout.

For those with dyslexia their difficulties were compounded by participants’ reluctance to disclose their disability or self-advocate for support for fear of discrimination, stigmatisation and retribution. This lack of perceived safety meant individuals had to apply their own strategies to manage workplace demands, leading to significant feelings of mental exhaustion. This concurs with reports by Nalavany et al. (2011) and McNulty (2003) who also noted substantial fatigue among adults with dyslexia as they managed the demands of work along with other competing life demands. The ripple effect of working longer hours to try to keep up with work demands led to some of our participants experiencing the additional impact of feeling isolated from their peers [77,78].

### Inadequate Job Resources

Workplaces are still largely designed to accommodate an abled population [37,79], and the results from our study suggest that adjustments and accommodations for people with dyslexia are not the norm. This echoes the findings of Deacon et al. (2020), who reported that when assistive technologies were not routinely offered, employees had to implement their own work-around strategies or procure their own assistive technologies to complete tasks. This is in the face of government legislation stating that adjustments should be applied to support workers with disabilities [22,25,79,80,81].

An important issue in the workplace is the ability for one to disclose their disability. Australian research by Wissell et al. (2020) found reasonably high rates of disclosure with 60% among (n = 65) participants were likely to disclose to a colleague at work. Evidence has found that for individuals to disclose, they must feel psychologically safe [68,69,80,81]. We cannot infer how psychologically safe participants in this study felt from the data collected, but it is notable that many had chosen not to disclose for fear of discrimination and/or shame, or because they felt their internal workplace policies were not adequately sensitive to the unique experiences of employees with dyslexia.

In the right environment, disclosure can be a positive process. It may facilitate a reduction in feelings of isolation, as being ‘out’ can facilitate social networks with others who can provide support to the employee and the employer [37]. It has also been found that self-disclosure may potentially reduce the stress related to concealing one’s identity [37,80,82,83,84].

Perceived limits to workplace progression were raised by several of our participants and this is unsurprising given their descriptions of burden, overwork, and fear of failure. Our findings align with MacDonald and Deacon (2019) who found that 44.4% of their cohort felt they had missed out on promotion opportunities due to their dyslexia. As positions advance, increased emphasis is often placed on organisational skills and written communication [34]; tasks that can be more difficult for employees with dyslexia. These limitations to career progression opportunities made some of our cohort feel trapped within their current roles or question their chosen career paths.

While many years have passed since the enactment of the Fair Work Act 2009 and the Disability Discrimination Act (DDA) 1992 [22,85], out participants maintained there was a lack of employer knowledge about their legal responsibilities in relation to supporting employees with disability in the workplace. The DDA was designed to prohibit discrimination against individuals with disabilities in employment, education, and in accessing premises, goods, and facilities [86]. However, our findings, along with those from other authors, suggest that there is still much to be done in ensuring that people with dyslexia do not feel stigmatised [33,69,87]. Australia must follow in the footsteps of countries such as the UK and US, where regulations are enforced to ensure people with dyslexia are properly protected under the law. This goes further towards ensuring employees with dyslexia can work to their full potential without fear of stigmatisation and discrimination.

## 5. Recommendations

### 5.1. Provide Supportive Job Resources

A review of workplace policies and procedures is needed to ensure (i) that recruitment practices are fair, reasonable and explicit in the type of support available to employees with learning disabilities [88,89]; and (ii) that employees with dyslexia and other learning disabilities clearly understand what supports are available to help them achieve their work goals.

Additionally, organisational training made available, especially for leaders could help develop understanding and awareness of dyslexia and create more inclusive workplace that may benefit dyslexic workers. This could include establishing avenues for peer support, building a psychologically safe environment for disclosure of disability and ongoing inclusion of people with disability, and building an understanding of what reasonable work adjustments are [29,30,88,90,91,92,93,94]. For example, when employers are cognisant that dyslexia can cause an employee to experience fatigue over and above someone without dyslexia, they may feel more equipped to provide appropriate job resources such as flexible working hours, additional breaks, and additional time to complete tasks.

Finally, strong working relationships and positive social interactions with peers, supervisors and managers is pivotal to the success of individuals with dyslexia and their ability to undertake help seeking and self- advocacy behaviours [95]. It is important for organisations to foster and encourage collegial relationships, where employees feel valued and accepted. High quality relationship with colleagues and one’s supervisor may alleviate the negative impacts of job demands, reduce the risk of job burnout, and improve workplace engagement [96].

### 5.2. Build a Culture That Fosters Autonomy

Our research brought to light that a work culture that enables individuals the opportunities to develop self-awareness, utilise their personal resources, and access the job resources they need for their roles fosters career satisfaction and leads to greater role autonomy. Job autonomy and work self-efficacy are vital for the health of employees, as it offers more opportunities to deal with high pressure situations [97], especially for people with dyslexia [13]. We found when participants had job autonomy, they described feeling high levels of self-efficacy, were less likely to feel job stress, workplace stress and job burnout, and were more likely to manage the day-to-day challenges of dyslexia.

It appeared that participants have high levels of self-efficacy and were also more likely to draw on their own personal resources, such as accessing external support or using personal software programs to assist them at work. This helped some to build a small sense of autonomy and control over their work environment even in cases where little or no job resources were provided.

### 5.3. Moderate Job Demands

Organisations could also focus on identifying job demands that are more likely to overwhelm the employee with dyslexia, leading to potential burnout. In this study, participants presented high levels of work engagement, but also significant levels of job burnout. Burnout can be moderated by various factors including being given greater autonomy at work, having supportive colleagues and managers, organisations taking steps to allow reasonable work adjustments [30], and employer training for more informed leadership teams [1,24].

## 6. Limitations and Future Directions

This study had several limitations. Our sample size was small, therefore the transferability of the findings in this study would need to be treated with caution. The personal experiences and knowledge of the interviewer, who has dyslexia, may have shaped data collection, although every attempt was made to counteract this through the prices of reflexive thinking and the close collaboration of the research team. There was a dependence on social media to enlist participants, which may have restricted the number of individuals who saw the recruitment flyer. Finally, of note, usually those with dyslexia will only have one assessment to diagnose their disability and then will not need to have any further assessments. However, we did not formally validate if participants had a formal diagnosis of dyslexia.

While our findings suggest an association between dyslexia, job demands, lack of job resources, and poor mental health, our methodology does not enable us to determine the strength of this association or confirm a causal relationship. Future research should consider using the JD-R Model within a quantitative case–control study to compare dyslexic and non-dyslexic employees.

## 7. Conclusions

This research explored the lived experiences of Australian adults with dyslexia in the employment context. Our work aligns with earlier conclusions in the general working population indicating that a rise in job demands reduces individual’s perceptions of their ability to complete work tasks [98,99,100,101,102]. We found that our 14 participants experienced a myriad of challenges across their employment histories. The result from this sample attributed many of these challenges to a lack of, or limited, awareness about dyslexia as a disability among managers, employers, human resource personnel, and colleagues, and being subject to negative perceptions, stereotypes, misunderstandings or discrimination. The JD-R Model proposes that excessive job demands, in the absence of supportive job resources and personal resources leads to poor mental health and wellbeing. Although preliminary, our results seem to suggest that employees with dyslexia face challenges in the workplace related to their disability including excessive mental exhaustion, and fatigue, leaving them vulnerable to workplace stress and job burnout.

Improving psycho-social workplace environments, increasing job resources, decreasing job demands, and critically influencing work engagement, will reduce job burnout and reduce apparent difficulties for individuals with dyslexia in the workplace.

In conclusion, even though this study was relatively small, it contributes to the expanding body of evidence regarding the challenges dyslexic employees face in Australian workplaces [30,31,34]. Employers play a vital role in mobilising job resources (e.g., social support, policies and procedures and increase self-efficacy) and avoid overwhelming job demands (e.g., work overload, and access to reasonable adjustments), to ensure those with dyslexia stay engaged and mentally healthy at work. Ensuring as well, organisational wide strategies that aim to improve engagement for those with disabilities such as dyslexia, and by developing more inclusive workplace cultures, as per Australian legislative requirements. This will likely result in increased productivity, which will ultimately benefit both employees and employers.

## Figures and Tables

**Table 1 ijerph-19-10719-t001:** Participant characteristics.

Gender	Age Range	State	Diagnosis Age Range	Professional Who Diagnosed	Education Level	Occupation	Employment Status	Industry
M	50–59	TAS	26–30	Psychologist	Postgraduate	Dean of University	Full time	Higher Education
F	50–59	NSW	5–12	Educational Psychologist	Postgraduate	Paediatric Nurse Manager	Full time	Government
M	30–39	WA	13–19	Educational Psychologist	Diploma	Mechanic	Full time	Mining
F	40–49	NSW	5–12	Psychologist	Diploma	Project Officer	Part time	Education
F	20–29	VIC	20–25	Provisional Psychologist	Diploma	Call Centre Operator	Full time	Call Centre
F	30–39	NSW	13–19	Psychologist	Postgraduate	Allied Health Assistant	Unassigned	Health Care
M	50–59	QLD	20–25	Educational Psychologist	Undergraduate	Project Officer	Full time	State Government
M	30–39	ACT	36–40	Neuropsychologist	Diploma	Tour guide	Casual	Hospitality and Tourism
F	50–59	QLD	46–50	Educational Psychologist	Year 12		Unemployed	Banking
F	40–49	VIC	5–12	Educational Psychologist	Postgraduate	Marketing and Communication Manager	Part time	Marketing
M	30–39	QLD	13–19	Unsure	Postgraduate	Unassigned	Full time	Government
F	30–39	VIC	20–25	Psychologist	Postgraduate	Senior Finance Manager	Full time	Mining
M	60–69	VIC	5–12	Other	Postgraduate	Professor Higher Education	Retired	Education
F	40–49	VIC	5–12	Unknown	Diploma	Disability Support Worker	Full time	Disability

## Data Availability

The data that support the findings of this study are available from the corresponding author upon satisfactory request.

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
