# Peer review of "“You Don’t Look Dyslexic”: Using the Job Demands—Resource Model of Burnout to Explore Employment Experiences of Australian Adults with Dyslexia"

_ijerph, 2022, doi:10.3390/ijerph191710719_

Round 1

Reviewer 1 Report

The purpose of this manuscript was to investigate the relationship between dyslexia and employee well-being.

The authors conducted an excellent literature review, and the work is well-presented and discussed, but I have some reservations about the experimental design,  but have some suggestions for improvement.

1. The literature review in the introduction must help strengthen the methodology employed in this paper. The reader should be persuaded and convinced of your decision to use this particular methodology in your research.
2. The previous suggestion also applies to the final paragraph of the introduction, where the purpose of this work is presented in a bold and unsupported statement: "We are using the model as a benchmarking instrument."
The sample size is small (n=14), and it is unclear why the authors would choose to use it.  Uncertainty surrounds the method for validating dyslexia in the participants. For instance, did they possess a recently issued certificate?
4. I was unable to discern how the sample's sector varied or did not vary; comparing burnout scores from different industries is difficult. This limitation eliminates the scope for a substantive discussion of the results.
5. In lines 386-389, the authors discuss the issue of several employees not disclosing their dyslexia, which makes me wonder who among the participants complained about a lack of "disability-appropriate job resources" (line 406). Those who did or those who did not disclose it?

6. If revised, this work should be formatted as a case report.

Author Response

Response to Reviewer 1 Comments

Point 1: The literature review in the introduction must help strengthen the methodology employed in this paper. The reader should be persuaded and convinced of your decision to use this particular methodology in your research.

Response 1: Thank you for this feedback. In response, we have reworked the literature review to strengthen the methods section and added to the methods section as well with the following:

Literature review

There is scarce theory-driven research and a significant need in Australia to increase knowledge regarding factors in the work environment that enhance or hamper occupational health and satisfaction, from the perspective of employees with dyslexia. Therefore, in this study, we wished to explore and better understand the working environment for people with dyslexia. To do this we have adopted the Job Demands–Resources Concept (JD-R), proposed by Demerouti et al. [56] as a theory- driven framework.

The Job Demands-Resource Model of Burnout (JD-R Model) (Appendix 1) is a validated tool that has been used to evaluate organisational outcomes across several workplace settings [39-41] and has generic application for all workplaces and their varying composition of employees. Currently there is only a small body of research using the JD-R Model on populations with disabilities [42-44], however not on a disability population such as dyslexia. For those with dyslexia it is not clear if, how and why workplace job resources and job demands can be connected to work‐related outcomes (such as job burnout). In the absence of a theory-driven workplace model within Australia we have adopt the JD-R Model to help better understand the link between dyslexia, “workplace characteristics” assesses job demands, job resources and personal resources and “employee wellbeing” explores worker burnout and engagement.

The aim of this research is to investigate the working experiences of adults with dyslexia by explore their organisation’s workplace practices and to derive suitable recommendations to support work participation.

Methodological framework:

We have not used the JD-R Model to compare whether our population are facing different challenges to the general population. Instead, the aim was to support the development of our categorisation framework and to utilise a theoretical framework to better understand and examine the association between job demands and employee wellbeing and motivation, and to explore how job demands may be mitigated by job or personal resources (occupational support/adjustments or personal characteristics/strategies).

Point 2: The previous suggestion also applies to the final paragraph of the introduction, where the purpose of this work is presented in a bold and unsupported statement: "We are using the model as a benchmarking instrument."

Response 2: Thank you for this feedback. In response, we have reworked the literature review to strengthen the methods section as outlined in response 1.

Point 3: The sample size is small (n=14), and it is unclear why the authors would choose to use it.  Uncertainty surrounds the method for validating dyslexia in the participants. For instance, did they possess a recently issued certificate?

Response 3: Sample size has been addressed in the methodological framework as outlined below:

 As there is little research exploring the associations between workplace demands and employee wellbeing for people with dyslexia, we adopted a qualitative design utilising in-depth interviews to gather substantive responses from participants. We chose semi-structured interviews, utilising an interview guide with a set of pre-determined questions. Further follow-up and clarifying questions emerged during interviews with participants. Qualitative research does not specify a fixed amount of data to be collected but continues until saturation is reached and the available data tell a rich, complex and in-depth story about the phenomenon of interest. Compared to quantitative research, sample sizes for qualitative research are typically much smaller [45, 46]. Saturation of data occurs when no new insights can be gained; this is commonly achieved with minimum sample sizes of 12 for qualitative studies [46-48]. As a result, 14 participants were deemed sufficient for this qualitative analysis and the scope of this research.

Validating dyslexia in the participant has been addressed in the limitation and furture directions section as outlined below:

Finally, of note, usually those with dyslexia will only have one assessment to diagnosis their disability and then will not need to have any further assessments. However, we did not formally validate if participates had a formal diagnosis of dyslexia.

Point 4: I was unable to discern how the sample's sector varied or did not vary; comparing burnout scores from different industries is difficult. This limitation eliminates the scope for a substantive discussion of the results.

Response 4: In response, we have added a statement about the general population and a comparison to a disability study in the discussion and reworked the limitations as follows:

Discussion:

Similar to local [68, 69-72] and international research [39, 40, 61, 73, 74]) that has used the JD-R Model we also found that when employees are faced with high job demands and low job resources that they can face high levels of job burnout. However, we could not determine whether those with dyslexia faced additional job burnout because of their disability compared to the general population, or differences in job burnout based on specific industries.

Research undertaken by Lehmann (2020) found that an increase in Multiple Sclerosis (MS) work difficulties could lead to problems meeting job requirements and thus to a depletion of energy and burnout, as well as an increased likelihood of leaving the workplace. When MS‐ related work difficulties were integrated within the health‐impairing process, it’s believed that taxing job attributes increase perceived MS‐ related work difficulties [44]. In line with Lehmann’s work, we found for our participants that their dyslexia difficulties could lead to burnout and reduction in energy due to the difficulties of meeting work requirements. As dyslexia difficulties are embedded within the health-impairing process, demanding job characteristics are assumed to make dyslexia work more challenging for people.  This may indicate that those with disabilities are at greater risk of job burnout and further research is warranted to better understand whether people with dyslexia are experiencing the same workplace difficulties as those within the general population or if their perceived dyslexia difficulties add another layer of complexity leading to exacerbated job burnout.

Limitations:

Finally, of note, usually those with dyslexia will only have one assessment to diagnose their disability and then will not need to have any further assessments. However, we did not formally validate if participants had a formal diagnosis of dyslexia.  

While our findings suggest an association between dyslexia, job demands, lack of job resources, and poor mental health, our methodology does not enable us to determine the strength of this association or confirm a causal relationship. Future research should consider using the JD-R Model within a quantitative case-control study to compare dyslexic and non-dyslexic employees.

Point 5. In lines 386-389, the authors discuss the issue of several employees not disclosing their dyslexia, which makes me wonder who among the participants complained about a lack of "disability-appropriate job resources" (line 406). Those who did or those who did not disclose it?

Response 5: This has now been amended to say ‘all participants’.

All the participants typically described a lack of disability-appropriate job resources (the physical, social or organisational factors that help an individual achieve their work goals and reduce stress) to support them during different points in their careers.

Point 6. If revised, this work should be formatted as a case report.

Response 6: Thank you for your suggestion, we have asked for editorial guidance as the team believe an article is the correct format for this work.

Reviewer 2 Report

The authors make a valuable contribution in linking two topics: dyslexia and work-related halth. The idea of using the JD-R model to explore the experiences of affected individuals in their workplace is original and promising.

In their text, the authors use key concepts of the model to show how (unfavorable) job demands and (lack of) resources are related to well-being (burnout). Maybe the focus on this negative relationship could possibly be mentioned in the title.

The material used and the explorative method are well suited. The comments on recruitment are not superfluous, but contribute to a better understanding of the situation of people with dyslexia. The references to possible improvement measures are very helpful because they open the perspective of a positive development in the future.

Overall, authors should keep in mind that the JD-R model nowadays often serves as a container for all kinds of content. They should resist to take only what may seem useful for them to say what they wanted to say anyway. From an occupational health perspective they might e.g. reflect on whether there is a special model emerging for people with dyslexia, or whether people with dyslexia are just 'normal cases' from the point of view of a general model.

It may be very exciting to see how dyslexic people could be detected as a certain group in a quantitative case-control study.

Author Response

Point 1: The authors make a valuable contribution in linking two topics: dyslexia and work-related health. The idea of using the JD-R model to explore the experiences of affected individuals in their workplace is original and promising.

Response 1: Thank you for your encouraging feedback on the overall paper.

Point 2: In their text, the authors use key concepts of the model to show how (unfavorable) job demands and (lack of) resources are related to well-being (burnout). Maybe the focus on this negative relationship could possibly be mentioned in the title.

Response 2: Thank you we have reviewed the title and amended it to the following:

You don’t look dyslexic”: Using the Job Demands - Resource Model of Burnout to explore employment experiences of Australian adults with dyslexia 

Point 3: The material used, and the explorative method are well suited. The comments on recruitment are not superfluous but contribute to a better understanding of the situation of people with dyslexia. The references to possible improvement measures are very helpful because they open the perspective of a positive development in the future.

Response 3: Thank you for your positive critique of the methods and improvements section we are pleased you feel it will be helpful for future research developments.

Point 4: Overall, authors should keep in mind that the JD-R model nowadays often serves as a container for all kinds of content. They should resist to take only what may seem useful for them to say what they wanted to say anyway. From an occupational health perspective they might e.g. reflect on whether there is a special model emerging for people with dyslexia, or whether people with dyslexia are just 'normal cases' from the point of view of a general model.

Response 4: Thank you for your feedback. In response, we have added a comparison disability study in the discussion and reworked the limitations as follows:

Discussion:

Similar in both local [68, 69-72] and international research [39, 40, 61, 73, 74]) that has used the JD-R Model, we also found that when employees are faced with high job demands and low job resources that they can face high levels of job burnout. However, we could not determine whether those with dyslexia faced additional job burnout because of their disability compared to the general population, or differences in job burnout based on specific industries.

Research undertaken by Lehmann (2020) found that an increase in Multiple Sclerosis (MS) work difficulties could lead to problems meeting job requirements and thus to a depletion of energy and burnout, as well as an increased likelihood of leaving the workplace. When MS‐ related work difficulties were integrated within the health‐impairing process, it’s believed that taxing job attributes increase perceived MS‐ related work difficulties [44]. In line with Lehmann’s work, we found for our participants that their dyslexia difficulties could lead to burnout and reduction in energy due to the difficulties of meeting work requirements. As dyslexia difficulties are embedded within the health-impairing process, demanding job characteristics are assumed to make dyslexia work more challenging for people.  This may indicate that those with disabilities are at greater risk of job burnout and further research is warranted to better understand whether people with dyslexia are experiencing the same workplace difficulties as those within the general population or if their perceived dyslexia difficulties add another layer of complexity leading to exacerbated job burnout.

Point 5: It may be very exciting to see how dyslexic people could be detected as a certain group in a quantitative case-control study.

Response 5: Thank you for your suggestion we have included it within the limitations section:

While our findings suggest an association between dyslexia, job demands, lack of job resources, and poor mental health, our methodology does not enable us to determine the strength of this association or confirm a causal relationship. Future research should consider using the JD-R Model within a quantitative case-control study to compare dyslexic and non-dyslexic employees.

Round 2

Reviewer 1 Report

I read the revised version of the manuscript and am pleased with the changes and improvements that resulted in the manuscript's limitations and weaknesses being overcome.

  I only have a couple of suggestions for improving the text.

For the clarity shake,,

Consider replacing the sentence in lines 614-616 with this sentence: In conclusion, despite the fact that this study was relatively small, it contributes to the expanding body of evidence regarding the challenges dyslexic employees face in Australian workplaces.

Consider replacing the sentence in lines 622-623 with this sentence:

This will likely result in increased productivity, which will ultimately benefit both employees and employers.

Author Response

Thank you for your positive feedback and your final suggestions, which have been incorporated into the final paragraph of the conclusion. 
